

# AIUB-GRACE gravity field solutions for G3P: processing strategies and instrument parametrization

Neda Darbeheshti[1], Martin Lasser[1], Ulrich Meyer[1], Daniel Arnold[1], and Adrian Jäggi[1]

[1]Astronomical Institute, University of Bern, Sidlerstrasse 5, 3012 Bern, Switzerland

*Correspondence to:* Neda Darbeheshti (neda.darbeheshti@gmail.com)

**Abstract.** This paper discusses strategies to improve the GRACE monthly solutions computed at the Astronomical Institute of the University of Bern (AIUB) which are contributing to the Horizon 2020 project G3P - Global Gravity-based Groundwater Product. To improve the AIUB-GRACE gravity field solutions, we updated the use of the Level-1B observations and adapted the background models, and improved the processing strategies in terms of instrument screening and parametrization. We

used the latest Release 3 K-Band product (KBR) and star camera data (L1B RL03), and adopted the latest Release 6 of the Atmospheric and Ocean De-aliasing (AOD1B RL06) product. For the accelerometer parametrization, we used arc-wise full scale factor matrix and arc-wise third-order polynomial biases. The new accelerometer parametrization is effective to reduce noise over the oceans in gravity field solutions especially for the late years of the GRACE mission when the thermal control was switched off. In this paper, we show that the outliers in KBR antenna offset correction (AOC) are projected into the range-rate

residuals; therefore, we used the KBR AOC as the main source for outlier detection and eliminated the AOC above a threshold for all data before the gravity field processing.

## 1 Introduction

The Global Gravity-based Groundwater Product (G3P) is a collaborative Horizon 2020 project between twelve European insti-

tutions coordinated by the German Research Centre for Geosciences (GFZ, (Güntner et al., 2020)). One of the key objectives of the G3P project is to process Gravity Recovery And Climate Experiment (GRACE, Tapley et al. (2004)) and GRACE Follow-On (GRACE-FO, Tapley et al. (2019)) Level-1B instrument data. GRACE and GRACE-FO provide a unique type of Earth observation from space (Wahr et al., 1998), total water storage variations on the continents, which is essential to calculate the variations in groundwater storage by subtracting other compartments of water storage variations such as glaciers, snow, soil

moisture and surface water bodies derived from other Earth observation data or models.

The Astronomical Institute of the University of Bern (AIUB) is one of the GRACE/GRACE-FO analysis centers contributing to the G3P project. AIUB has produced monthly GRACE gravity field solutions since 2011. The first monthly GRACE series was released in 2011, (Meyer et al., 2012) and the second series in 2016, AIUB-RL02, (Meyer et al., 2016). Beside the GRACE





gravity field solutions, GPS-based GRACE orbits are also processed at AIUB and are made publicly available (Arnold and Jäggi, 2020). The AIUB-RL02 of monthly gravity field models contributed to the combined monthly models in the frame of the European Gravity Service for Improved Emergency Management (EGSIEM) project (Meyer et al., 2019) coordinated by AIUB (Jäggi et al., 2019). AIUB-RL02 lacks the late years of GRACE gravity field solutions in 2016 and 2017. Therefore a

new AIUB-G3P is prepared with the following objectives:

–  provide the complete time series of monthly GRACE gravity field solutions from 2002 to 2017.

–  update the input observations and background models for GRACE gravity field recovery with the latest available (see section 2).

–  improve gravity field recovery processing.

–  contribute to the International Association of Geodesy (IAG) service COST-G (International Combination Service for Time-variable Gravity Field Solutions)(Jäggi et al., 2020).

–  contribute to the Global Gravity-based Groundwater Product (G3P).

## 2  GRACE orbit dynamic model

AIUB-G3P, like its two predecessors AIUB-RL01 (Meyer et al., 2012) and AIUB-RL02 (Meyer et al., 2016), is based on

the Celestial Mechanics Approach (CMA) (Beutler et al., 2010), which treats gravity field estimation as a generalized orbit determination problem. The equations of motion for both GRACE satellites are:

$$\ddot{\boldsymbol{r}} = \boldsymbol{a}_g + \boldsymbol{a}_{ng} + \boldsymbol{a}_{emp} \tag{1}$$

where $\ddot{\boldsymbol{r}}$ is the acceleration of the satellite, second time derivative of the satellite position vector, $\boldsymbol{a}_g$ denotes accelerations due to all gravitational forces, $\boldsymbol{a}_{ng}$ denotes accelerations due to all non-gravitational forces and $\boldsymbol{a}_{emp}$ denotes empirical accelerations

designed to overcome deficiencies remaining in the force models.

The CMA solves equation (1) as a linearized least-squares estimation where gravity field coefficients and all other orbit related parameters are estimated together. Kinematic positions and K-Band range-rate data are used as observables to estimate orbit and gravity field coefficients such that the orbit trajectories are solving equation (1).

The gravitational models ($\boldsymbol{a}_g$) in equation (1) are called the background gravity models. The details of the background

gravity models for AIUB-G3P are provided in Table 1.

For $\boldsymbol{a}_{emp}$, constrained piecewise constant accelerations at 15 minutes intervals in all three directions of the local orbital frame were estimated (Jäggi et al., 2006).

For $\boldsymbol{a}_{ng}$, the GRACE accelerometer data is used. The accelerometer measurements given in the ACC1B data product, are affected by unknown scale factors, biases and random noise (Kim, 2000). Ideally, the scale factor matrix $\mathbf{S}$ (equation 2) should





**Table 1.** Background models for AIUB-G3P with maximum spherical harmonic degree (d/o) (if applicable).

| Model | Description | Reference |
|---|---|---|
| A priori gravity | AIUB-GRACE03S (static part), d/o 160 | |
| Solid Earth tides | IERS 2010 conventions | (Petit and Luzum, 2010) |
| Ocean tides | FES2014b, d/o 100 | (Carrere et al., 2016) |
| Atmosphere and oceanic variability | AOD1BRL06, d/o 100 | (Dobslaw et al., 2017) |
| Solid Earth pole tide | IERS 2010 conventions | (Petit and Luzum, 2010) |
| Ocean pole tide | IERS 2010 conventions, d/o 100 | (Desai, 2002) |
| N-Body perturbations | DE421 | (Folkner et al., 2009) |

be an identity matrix, but it contains non-unit diagonal elements and non-zero off-diagonal elements due to small instrument imperfections causing mutual influence of the accelerometer axes among each other. In order to account for these imperfections, a fully-populated scale factor matrix is used for AIUB-G3P:

$$\mathbf{S} = \begin{bmatrix} s_x & \alpha + \zeta & \beta - \epsilon \\ \alpha - \zeta & s_y & \gamma + \delta \\ \beta + \epsilon & \gamma - \delta & s_z \end{bmatrix} \tag{2}$$

The off-diagonal components are composed of a symmetric shear $\alpha$, $\beta$ and $\gamma$ and a skew- symmetric rotation part $\zeta$, $\epsilon$ and $\delta$. For more details on interpretation of these elements we refer to (Klinger and Mayer Gürr, 2016). For previous AIUB releases, the off-diagonal elements have been neglected, i.e. the scale factor matrix was assumed to have main diagonal elements only. To account for instrument imperfections and misalignment, for AIUB-G3P both main diagonal and off-diagonal elements of the scale factor matrix (cf. Eq. 2) were estimated on a daily basis.

To account for bias changes due to temperature variations according to (Klinger and Mayer Gürr, 2016), a bias vector $\boldsymbol{b}$ is estimated daily using a third order polynomial. Four coefficients were estimated in each direction, therefore twelve accelerometer bias coefficients were estimated on a daily basis for each satellite.

The main observations are a combination of kinematic orbit positions for each satellite and inter-satellite K-Band range-rate measurements. The combination is realized through daily normal equations and is accumulated for one month to solve for

monthly spherical harmonics coefficients of the Earth's gravity field. The kinematic orbits of the GRACE satellites are determined in a precise point positioning from the undifferenced GPS phase observations (Jäggi et al., 2006). The kinematic orbits rely on reprocessed GPS orbits from the CODE analysis centre (Steigenberger et al., 2011). For kinematic orbit determination maps of the empirical phase center variation of the GRACE GPS antennas (Jäggi et al., 2009) were re-estimated. The kinematic orbits for the whole GRACE lifetime can be downloaded from http://ftp.aiub.unibe.ch/LEO_ORBITS/GRACE/RL01/.

The specifics of the GRACE data products are given in the Table 2.



**Table 2.** Data products for AIUB-RL03

| Product ID | Release | Data Rate |
|---|---|---|
| KBR1B | RL03 | 5-second range-rate |
| KIN | AIUB in-house | 30-second position |
| ACC1B | RL02 | 1 second linear acceleration |
| SCA1B | RL03 | 1 second quaternion |

Although we can not see the star camera data in the equation (1) directly, they appear implicitly in two ways in GRACE gravity field recovery:

- transforming the linear ACC1B product to inertial frame (Darbeheshti et al., 2017),

- calculating KBR antenna offset correction in KBR1B product.

5  The SCA1B product is used to define the KBR antenna offset correction (AOC) in the KBR1B product. The KBR instrument measures the distance between the antenna phase centers, which are placed nominally on the satellite frame x-axis, almost 1.5 m away from the satellites' center of mass. Consequently, any pointing jitter (deviations of the satellites' attitudes from their nominal attitudes) causes a geometric error in the ranging measurement. In the absence of such misplacements and in the absence of pointing jitter, this effect would be constant and hence not effect the measured (biased) KBR range. The GRACE

10  KBR1B data product files contain a column, which is called antenna offset correction (AOC) term (Case et al., 2010). It has to be added to the KBR ranging measurement. A second and third column is also provided, computed by numerical differentiation, describing the correction for range-rate and range-acceleration.

Although the AOC is improved in GRACE KBR1B RL03, AOC outliers exist and need to be removed. AOC rate is in the range of $\pm 0.5 \mu m/s$ (Klinger, 2018), therefore the values beyond $\pm 1 \mu m/s$ are considered outliers. Figure 1 shows range-rate

15  AOC columns from GRACE KBR1B RL02 and GRACE KBR1B RL03 for two days in 2006. There are not any outliers in the day 264, but for both RL02 and RL03, day 290 contains outliers. The corresponding amplitude spectral density (ASD) plot of these two days show although the high frequency noise (greater than $10^{-2} \mu m/s/\sqrt{Hz}$) has been filtered out from the AOC, AOC RL03 still contains outliers that correspond to the satellite events like calibration maneuvers for different instruments.





**Figure 1.** Range-rate AOC for days 264 (no outliers) and 290 (containing outliers) with the corresponding ASD.

## 3    Pre-processing: Level 1B data Screening

In theory, the GRACE L1B data products can be used directly for gravity field recovery, but there are outliers in the data that need to be removed before the gravity field recovery processing. Data screening for 15 years of GRACE data and for every instrument is a challenging task. The cause of systematic errors and outliers has been studied by (Goswami, 2018) and (Klinger, 2018). In this work we have only focused on finding an effective way to find outliers in GRACE data.

The overall error (including outliers) in the instrument data and background models are projected in range-rate residuals in case of GRACE gravity field recovery. Figure 2 shows how outliers in range-rate AOC are mapped into range-rate pre-fit residuals.



**Figure 2.** Outliers in AOC (day 80, 2003) are mapped into the KBR range-rate pre-fit residuals.

**AOC and ACC Screening**

AIUB-RL02 screening was based on inspection of the range-rate residuals and removing outliers by gap tables (Meyer et al., 2016). For AIUB-G3P, we developed a new screening strategy. Our new strategy for AIUB-RL03 is inspecting the GRACE L1B data product itself, and whenever we find out an outlier, we throw out the epoch of the outlier by means of monthly session

5    tables in the Bernese software. AIUB GRACE gravity field solutions are based on daily arcs. This means we estimate orbital





parameters for each 24 hours arc. The epochs of these daily arcs are kept in the session tables. When we find an outlier, by removing the outlier epoch, we split the daily arc and estimate the orbital parameters for new arcs. Therefore, in general AIUB monthly gravity solution is based on daily arcs, but in months that there are outliers in the instrument, there are shorter arcs in the monthly gravity solution.

We performed the data screening in three major steps: (1) threshold-based outlier detection of KBR1B AOC rate data product, (2) threshold-based outlier detection of ACC1B data product, and (3) empirical elimination of days that degraded final monthly gravity field solution. Table 3 summarizes the threshold values and margins used for threshold-based outlier detection for first and second steps. Margin means the time span before and after an outlier detected in the data. We used absolute value thresholds for the AOC rate data product, because the AOC rate is in the range of $\pm 0.5 \mu m/s$. For ACC1B data product, we

used daily median based threshold, because we estimate ACC scale factors and biases on daily basis.

**Table 3.** Level-1B data screening: Threshold-based outlier detection.

| Data product | Data type | Threshold | | Margin |
|---|---|---|---|---|
| KBR1B | antenna offset correction (range-rate) | $|AOC_{\dot{\rho}}^{i}| < 1$ | $\mu m/s$ | 10 minutes |
| ACC1B | linear acceleration | $|a_x^i - \tilde{a}_x| < 10$ | $\mu m/s^2$ | 10 minutes |
| | | $|a_y^i - \tilde{a}_y| < 10$ | $\mu m/s^2$ | 10 minutes |
| | | $|a_z^i - \tilde{a}_z| < 10$ | $\mu m/s^2$ | 10 minutes |
| | | $\tilde{a}$ is the median of a day | | |

Figure 3 shows the periods of outliers in the AOC rate and linear ACC data for GRACE A and GRACE B in all three axes. The years 2006 and 2007 are representative of high quality of GRACE data. There are complete twelve months of instrument data products from 2003 to 2010, therefor there is not any data gap in the AOC rate and linear ACC data for years 2006 and 2007. GRACE data products are only available until end of June 2017. Since April 2011, the onboard instruments are shut

down for approximately 40-50 days during each 161 day to extend GRACE batteries's life time (Tapley et al., 2015), that's why there are AOC rate and linear ACC data gaps for 2016 and 2017, the last years of GRACE lifetime, where the quality of GRACE instrument data products is degraded.

The first column of figure 3 shows that the AOC outlier time length in 2007 is more than twice as 2006 (4 hours in 2006 versus 10 hours in 2007). In general, a satellite event like a calibration maneuver causes outliers in AOC. GRACE satellite

events are published in the Sequence of Events file. For 2017, a much looser outlier threshold ($3 \mu m/s$) had to be used for AOC, because with a $1 \mu m/s$ AOC threshold, we would not have any data left to solve for the gravity field recovery.

Columns two and three of figure 3 show that there are much less outliers in ACC data than AOC. GRACE A ACC data for the years 2006 and 2007 is of high quality, but there are some outliers in the GRACE B ACC data. In October 2016, the accelerometer on-board GRACE B was permanently powered-off to reduce the stress on the remaining battery cells. Since

then, no GRACE B accelerometer data is available (except for May 2017). To allow for gravity field recovery, the GRACE B



accelerometer transplant data (Bandikova et al., 2019) have been made available. 3 shows that the bias along y axis in GRACE B in November 2016 suddenly changes and follows the bias in GRACE A, this pattern continues in 2017, except in May, when real GRACE B ACC data is again available. In June 2017 the bias in GRACE B y axis is again the same as for GRACE A.



**Figure 3.** Annual plots of AOC rate (first column) and ACC1B accelerometer linear accelerations along three axes for 2006, 2007, 2016 and 2017. Red vertical lines in first column are outliers in AOC. For linear accelerations, vertical lines are outliers along three axes.



**Empirical elimination of whole days**

Empirical elimination of whole days has been used at the last stage of data screening for the generation of the AIUB-RL02 and AIUB-RL01solutions. For the empirical elimination procedure, $n$ monthly gravity field solutions are produced for each month, while $n$ is the number of days in each month. In each gravity field solution one whole day data is eliminated. Then

5   by plotting and comparing the $n$ gravity field solutions in terms of difference degree amplitudes of geoid heights, days that corrupt the monthly gravity solution may be recognized. Figure 4 demonstrates this procedure for July 2011. For this month, there are 27 days of data available. Therefore for iteration 1, there are 27 gravity solutions, where for each solution, one day has been eliminated (and consequently just 26 days have been used). Then for the second iteration, we have eliminated day 186. Consequently, the second iteration has been done with 26 gravity solutions. In the third iteration day 187 and in the fourth

10  iteration day 188 have been eliminated. In the 4th iteration, after eliminating three whole days, all the gravity solutions are converged and they are very close to each other and they are even of better quality than AIUB-RL02 monthly gravity field solution.

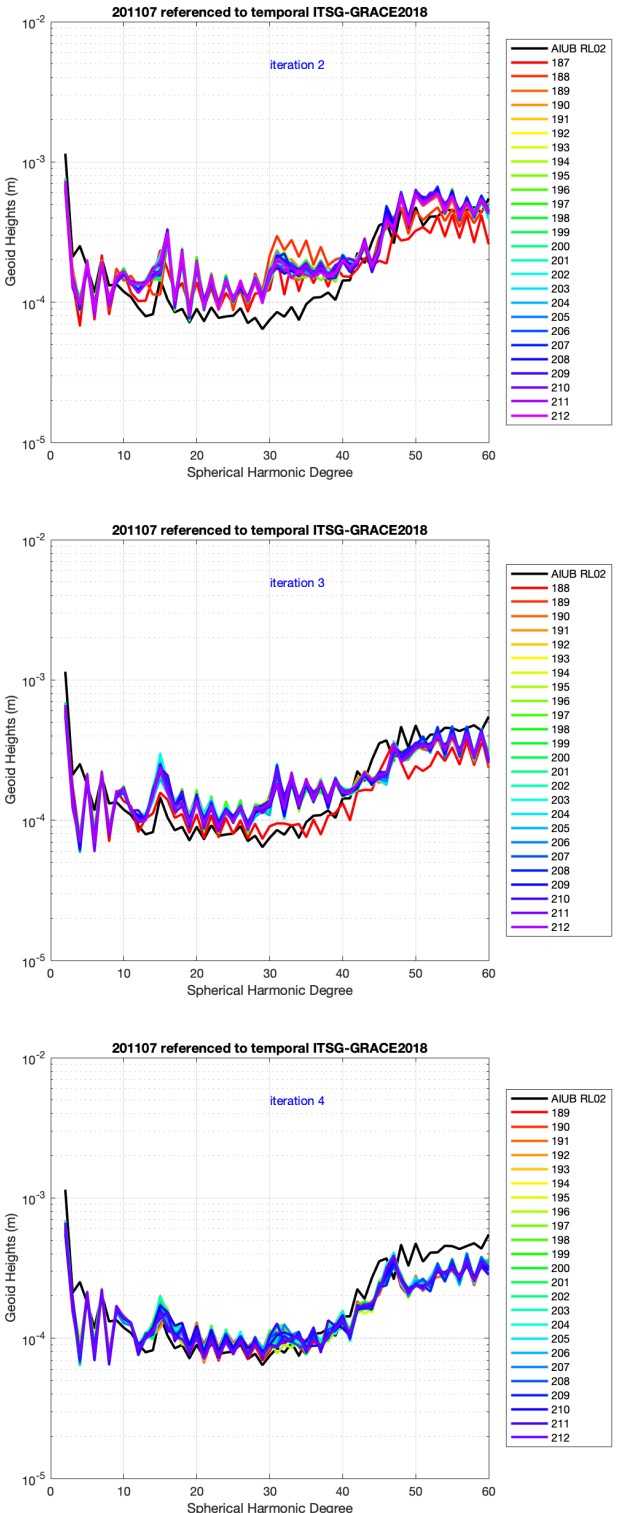

**Figure 4.** Empirical elimination procedure for July 2011, where in each iteration one day has been eliminated. In iteration 2, day 187 (red curve), and in iteration 3 day 188 (red curve) are eliminated. Red curves for these two iterations are clearly below other gravity solutions, which indicates, day 187 and 188 are corrupting the gravity solution. The legend shows the day number that has been eliminated in the gravity field solution.

To diagnose why these three days corrupt the gravity field solution, it is helpful to look at pre-fit residuals (Darbeheshti et al., 2018) of observations. Pre-fit residuals in context of gravity field recovery are observed value minus computed value, where the computed value is independent of gravity field estimation. Figure 5 shows daily root mean square (RMS) of pre-fit residuals for GRACE A and GRACE B orbits, ranges and range-rates in July 2011. Days 186, 187 and 188 show large RMS for all pre-fit residuals, which is in agreement with the empirical elimination procedure.

**Figure 5.** Daily pre-fit residual RMS for GRACE A orbit (radial (red), along- rack(green), cross-track(blue) directions, GRACE B orbit, range and range-rates for July 2011.



## 4 Evaluation of new AIUB-G3P GRACE

I this section, we compare new monthly GRACE solutions for G3P project, AIUB-G3P to AIUB-RL02. An overall comparison of the AIUB-RL02 and AIUB-G3P is shown in Figure 6. For comparison, we only considered the months where AIUB-RL02 is available. The new AIUB-G3P shows the lower noise level, which is the result of the improvements in the processing chain.

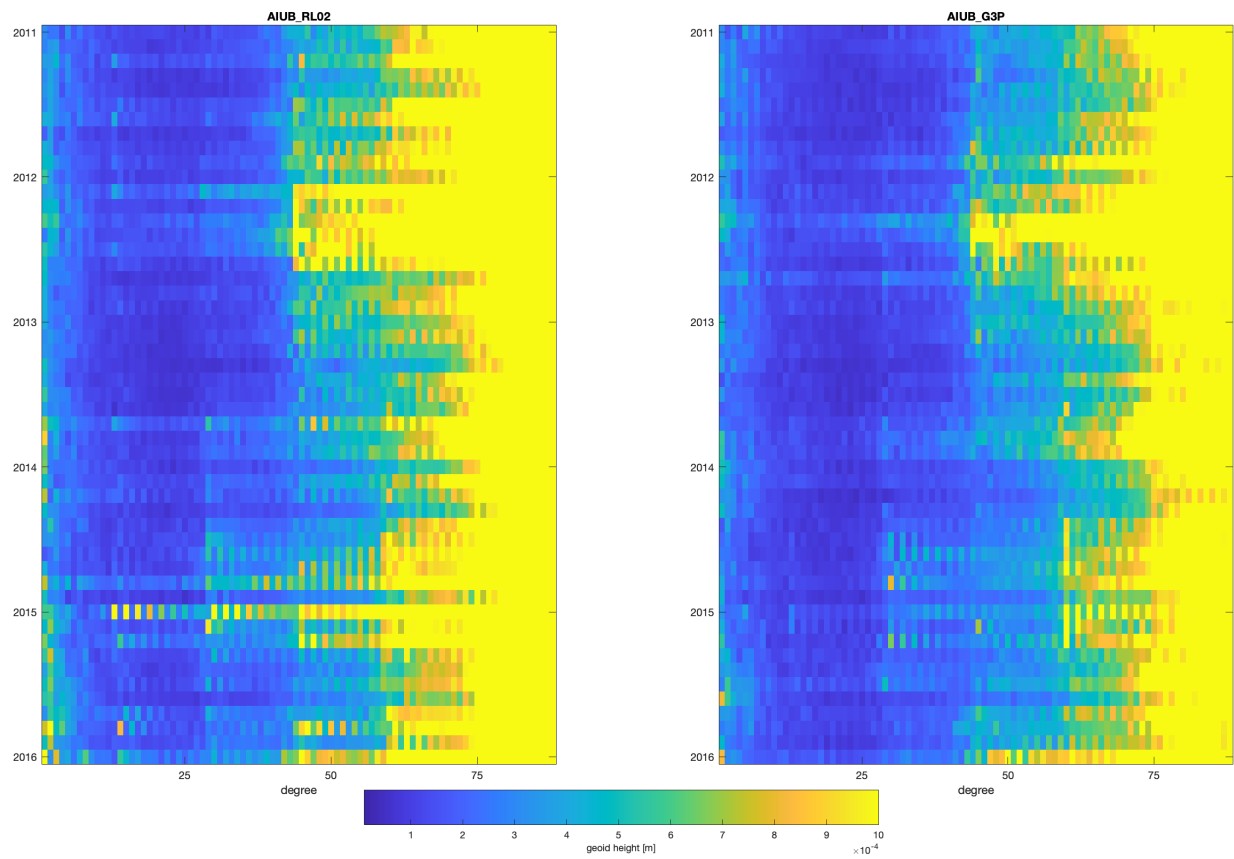

**Figure 6.** Difference degree amplitudes with respect to the 'mean model' for GRACE (February 2011 to August 2016) for AIUB-RL02 and AIUB-RL03.

5       One of the ways to assess GRACE gravity monthly solutions is to calculate the standard deviation (STD) of the variability over the oceans which should be weighted by the cosine of the latitude (Bonin and Tapley, 2012). This measure is called weighted STD over the ocean. We first subtract a 'mean model' from each gravity field solution and apply a 400 km Gaussian filter (Wahr et al., 1998), then we put an ocean mask on each monthly global gravity solution to calculate the STD over the oceans, which represent noise over the ocean for GRACE monthly gravity solution. The 'mean model' is the average of

10      monthly gravity field solutions provided by the Center for Space Research at the University of Texas, Austin (CSR Release 06),




the German Research Centre for Geosciences (GFZ Release 06), AIUB-RL02, the Centre National d'Etudes Spatiales/Groupe de Recherche de Geódeśie Spatiale (CNES_GRGS_RL04) and the Institute of Geodesy at Graz University of Technology (ITSG-Grace2018) for the time period 2004-2017. For detailed information about how the mean model is computed, see Peter et al. (2022).

5    Figure 7 shows the noise over the ocean for AIUB-RL02 and AIUB-G3P GRACE gravity field solutions. AIUB-G3P solution shows a significant improvement over AIUB-RL02 in the later years of GRACE lifetime, mainly after April 2011. There are few months in 2005 and 2009 for which the gravity field solutions are slightly worse in AIUB-G3P in terms of noise over the ocean. The reason for this degradation is still unknown to us. Figure 8 shows the RMS over the years spatially. The spatial plot clearly shows the overall improvement of AIUB-G3P compared with AIUB-RL02.

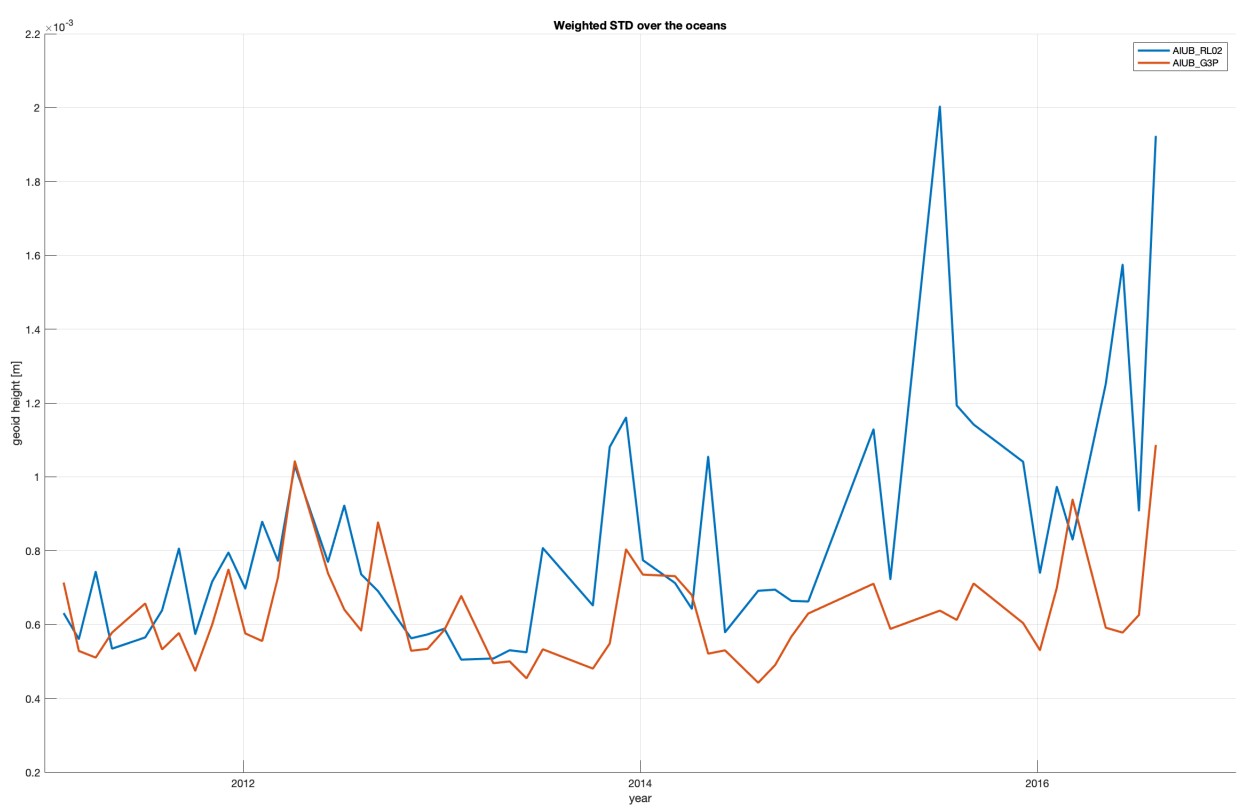

**Figure 7.** Noise over the oceans with respect to the 'mean model' for GRACE (February 2011 to August 2016) for AIUB-RL02 and AIUB-G3P.




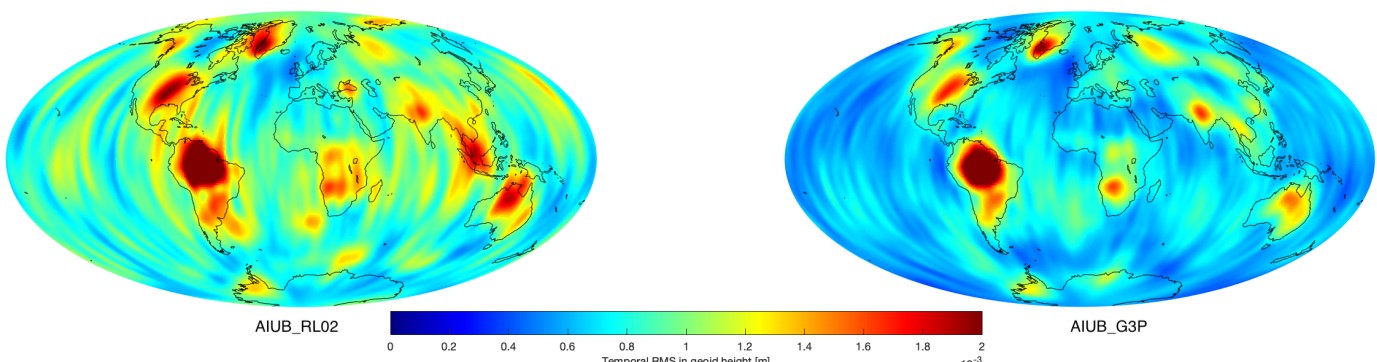

**Figure 8.** RMS anomalies with respect to the 'mean model' for GRACE (February 2011 to August 2016) for AIUB-RL02 and AIUB-G3P.

**Major improvements in the processing chain for AIUB-G3P**

Two major changes in the processing chain have contributed to the improvement of AIUB-G3P:

- Accelerometer parameterization: For AIUB-RL02, only the diagonal elements of scale factor matrix were calculated for each accelerometer for each arc. For AIUB-G3P, the full scale factor matrix was calculated for each accelerometer for
5   each arc, following the recommendations by Klinger et al. (2016).

- Updating AOD release from RL05 to RL06

The effect of these two changes is evaluated in Figure 9 for 2016-2017 when the quality of GRACE observations was degraded. AIUB-RL02 solution is based on AOD-RL05 and diagonal scale factor accelerometer parametrization, while AIUB-G3P is based on AOD-RL06 and full scale factor accelerometer parametrization. In the legend 'AOD-RL05' indicates the
10  same set up as AIUB-G3P, but using AOD-RL05 instead of AOD-RL06 and 'Diagonal' shows the same set up as AIUB-G3P, but diagonal scale factor accelerometer parametrization instead. Figure 9 shows the full scale factor matrix for accelerometer parametrization is important for late years of GRACE data. The effect of full scale factor accelerometer parametrization is even more important than updating to AOD-RL05, as it is clear from the noise over the ocean, the solution with AOD-RL05 is below the solution with diagonal scale factor accelerometer parametrization.

Earth System
**Science**
**Data**
Open Access Discussions



**Figure 9.** Noise over the oceans with respect to the 'mean model' for 2016 months where AIUB-RL02 are available. 'Diagonal' shows AIUB-G3P set up with diagonal scale factor accelerometer parametrization and 'AOD-RL05' shows AIUB-G3P set up with AOD-RL05.

Figure 10 shows the elements of accelerometer full scale factor matrix for AIUB-G3P GRACE B. The main-diagonal elements of the accelerometer cross-track axis are more scattered than elements of the along-track and radial axes, which is related to the smaller sensitivity of the cross-track axis compared to other two axes. Additionally, the shear and rotational elements associated with the less-sensitive cross-track axis ($\delta$ and $\gamma$) are non-zero and are increasing for the late years of GRACE. The





sheer and rotational elements are absorbing accelerometer imperfections and misalignments, resulting in the better quality of AIUB-G3P than AIUB-RL02 in figure 7.



**Figure 10.** Elements of the scale factor matrix for GRACE B (up) Main diagonal elements in along-track, cross-track and radial direction; for a better illustration, a constant offset of $\pm 1$ is added to the red and green graphs (middle) shear elements and (bottom) rotational elements.

Beside the official GRACE B transplant data provided by Jet Propulsion Laboratory (JPL) for the last six months of GRACE mission, the Institute of Geodesy at Graz University of Technology (TUG) also provides GRACE B transplant data (Behzadpour et al., 2021), which can be accessed via https://ifg.tugraz.at/downloads/gravity-field-models/alternative-grace-fo-11b/.

Figure 11 shows the gravity field solutions using GRACE B accelerometer transplant data from JPL and TUG. TUG ACT(blue curve) shows much better performance for November and December 2016 and April 2017. For January and June





2017, JPL and TUG gravity field solutions are similar. March 2017 gravity field solutions for both JPL and TUG has a poor quality, that's why it is plotted just up to degree and order 70.

**Figure 11.** Difference degree amplitudes in terms of geoid height with respect to GOCO05S solutions comparing GRACE B accelerometer transplant data from JPL and TUG.



## 5    Conclusions

In this paper, the importance of Level-1B data pre-processing methodologies to improve GRACE gravity field solutions were demonstrated on the basis of AIUB processing chain and the transition from AIUB-RL02 (Meyer et al., 2016) to AIUB-G3P. Also the contribution of individual updates to the overall accuracy improvement of AIUB-G3P was highlighted. In particular, the effects and benefits of an automated AOC data screening, and a full scale factor accelerometer parametrization were analyzed in detail. We also compared JPL and TUG accelerometer transplant data product. Overall, TUG accelerometer transplant data product perform better in terms of gravity field solution for last months of GRACE life time in 2016 and 2017. The full time series of GRACE AIUB-G3P gravity solutions can be accessed from the International Center for Global Earth Models website (ICGEM, (Ince et al., 2019)) at http://icgem.gfz-potsdam.de/series/03_other/AIUB/AIUB-G3P.

*Data availability.*

The full time series of GRACE AIUB-G3P gravity field solutions (Darbeheshti et al., 2023) is available at http://icgem.gfz-potsdam.de/series/03_other/AIUB/AIUB-G3P.



*Author contributions.* Adrian Jäggi reviewed the article. Ulrich Meyer reviewed the article. Martin Lasser developed accelerometer parametrization code. Daniel Arnold processed GRACE kinematic orbits. Neda Darbeheshti performed the GRACE AIUB-G3P gravity field processing and prepared the manuscript with contributions from all co-authors.

*Competing interests.* The authors declare that they have no conflict of interest.

5 *Acknowledgements.* This project is supported by funding from the " G3P - The Global Gravity-based Groundwater Product" by the European Union's Horizon 2020 research and innovation programme under grant agreement No 870353. Calculations in this study were performed on UBELIX, the HPC cluster at the University of Bern.



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
