# Peer review of "AIUB-GRACE gravity field solutions for G3P: processing strategies and instrument parametrization"

_Earth System Science Data, 2023_

## Author Comment (AC1)

Thank you for your valuable comments; we appreciate your insightful feedback. Here are the responses:

**RC1**: 'Review of essd-2023-72', Henryk Dobslaw, 04 Jul 2023
The manuscript „AIUB-GRACE gravity field solutions for G3P: processing strategies and instrument parametrizations" summarizes the analysis decisions taken at the Astronomical Institute of University of Bern (AIUB) for deriving monthly-mean gravity fields from sensor data collected by the GRACE and GRACE-FO satellite missions. Calculating gravity fields is a complex task involving the careful consideration of data from a range of different sensors (i.e., range-rate tracking, accelerometers, star cameras, GNSS navigation signals, etc.). Besides common standards & conventions, a number of choices are available to the processing centers like AIUB for individual aspects of the data pre-processing, screening, and integration. Documenting those choices as done in the current manuscript is important to foster international collaboration in order to identify the most promising ways to calculate GRACE-based gravity fields. The manuscript is generally well written and certainly fits the scope of the journal. Nevertheless, a number of comments might be considered before eventually accepting the article.

More details are needed on the outlier handling (P6L3). It seems to me that complete daily arcs are flagged and ignored, is that true at this point already? Do you have an idea about how much (potentially valid) data is removed by this step? What about options to calculate arcs that span over shorter periods of time, or that are shifted by 6 (or 12) hours?

The paragraph was revised:

For AIUB-G3P, a novel screening strategy has been developed. This approach involves scrutinizing the GRACE L1B data product, specifically KBR1B and ACC1B. When an outlier is identified in the daily KBR1B and ACC1B, the corresponding day is flagged. Subsequently, the epoch of the outlier is excluded using monthly session tables in the Bernese software.

The AIUB GRACE gravity field solutions are constructed by estimating orbital parameters for each 24-hour arc. The epochs of these daily arcs are recorded in the session tables. In the presence of an outlier, the affected epoch is removed, leading to the segmentation of the daily arc. New orbital parameters are then estimated for the revised arcs. As a result, while the general AIUB monthly gravity solution is typically based on daily arcs, months with outliers in the instrument exhibit shorter arcs in the monthly gravity solution.

The question above is also related to your statements on p10, where the elimination of whole days is described as the last stage of data screening. From the text, it is not immediately clear which approach exactly is taken for the most recent G3P release from AIUB. AIUB-RL01 does not need to be discussed again at this stage. If there are no changes with respect to AIUB-RL02, this should be stated so.

The sentence was revised:

The empirical elimination of entire days has been incorporated as the final stage of data screening in generating the AIUB-RL01, AIUB-RL02, and AIUB-G3P solutions.

Results presented in Figure 7 need further analysis, in particular with respect to the months that appear to be worse than in previous releases. I suggest to explicitly map (i) the monthly-mean background model, (ii) the mean from Peter et al. (2022) and the residual obtained with both AIUB releases leading to the RMS values presented in the time-series for those months in question. Please exclude continental signals and choose a color range & spatial smoothing that amplifies signals in the oceans.

This paragraph was added and the reference to Peter et al. 2022 was eliminated.

To maintain consistency, all comparisons in this section are referenced to a 'mean model'. The 'mean model' was computed by averaging monthly gravity field solutions from the Center for Space Research at the University of Texas, Austin (CSR Release 06), the German Research Centre for Geosciences (GFZ Release 06), the Centre National d'Etudes Spatiales/Groupe de Recherche de Ge\'ode\'sie Spatiale (CNES\_GRGS\_RL04) and the Institute of Geodesy at Graz University of Technology (ITSG-Grace2018) for the time period 2004-2017.

The paragraph was changed into:

One approach for evaluating GRACE gravity monthly solutions involves calculating the standard deviation (STD) of variability over the oceans, where hydrological signals are not expected. The discrepancies between the monthly solution and the 'mean model' are assessed on a grid with a cell size of 3 degrees, corresponding to a spherical harmonic expansion up to degree and order 60. Secular and seasonal variations are fitted to all grid cells and subtracted to eliminate long-periodic signals of oceanic origin. The grid cells are weighted by the cosine of the latitude to account for their different sizes, and the standard deviation over all ocean cells is computed. To prevent contamination from continental signals, the shoreline is shifted by three grid cells (equivalent to 9 degrees) into the oceans. Figure 7 shows the standard deviations computed in this way for AIUB-RL02 and AIUB-G3P GRACE gravity field solutions.

P1L5: RL06 is not the latest release of AOD1B anymore, since release 07 is already available (https://doi.org/10.1093/gji/ggad119). I suggest to skip the word „latest" and mention only that RL06 is now incorporated instead of RL05, which makes this processing choice consistent with the RL06 gravity fields of both the SDS and many other contributors to Cost-G.

The word "latest" was eliminated and the following sentence was added to the table 1.

update the input observations and background models for GRACE gravity field recovery to be consistent with the other contributors to the COST-G (see section 2).

P3L1: „non-unit" diagonal elements sounds odd. Please revise.

"non-unit" was eliminated.

P4L1: we can not see the star camera data in eq. 1 -> star camera data is not explicitly identified in eq. 1

The sentence was revised:

Although we can not see the star camera data in the equation (1) explicitly, they appear implicitly in two ways in GRACE gravity field recovery:

P21L1: Author contributions of AJ and UM might be elaborated a little further in view of, e.g., software heritage used for this work.

The sentence was revised:

Ulrich Meyer and Adrian J{\"a}ggi have made contributions to the software heritage utilized in this project and reviewed the article.

---

## Author Comment (AC2)

Thank you for your valuable comments; we appreciate your insightful feedback. Here are the responses:

**RC2**: 'Comment on essd-2023-72', Anonymous Referee #2, 20 Nov 2023
The Astronomical Institute, University of Bern is processing the GRACE/GRACE-FO data for many years and generates GRACE/GRACE-FO based temporal gravity field solutions and orbit products. Recently, AIUB published a new release of its gravity field time series called AIUB-G3P. This paper now describes the processing strategy for this new release, especially compared to that for the previous release AIUB-RL02. I appreciate this paper since it gives important hints and background information about AIUB's new gravity field product. Nevertheless, I'm not satisfied with some items, especially in the evaluation part, see my comments below. Therefore, I vote for major revision.

I have the following specific comments:

P6L3: You write: "Our new strategy for AIUB-RL03 is inspecting the GRACE L1B data product itself, and whenever we find out an outlier, we throw out the epoch of the outlier …" My recommendation: You should mention already here in this sentence in which kind of data you are looking for outliers. I suggest to write "Our new strategy for AIUB-RL03 is inspecting the GRACE L1B data product itself (KBR1B and ACC1B), and whenever we find out an outlier, we throw out the epoch of the outlier …"

The paragraph was revised:

For AIUB-G3P, a novel screening strategy has been developed. This approach involves scrutinizing the GRACE L1B data product, specifically KBR1B and ACC1B. When an outlier is identified in the daily KBR1B and ACC1B, the corresponding day is flagged. Subsequently, the epoch of the outlier is excluded using monthly session tables in the Bernese software.

The AIUB GRACE gravity field solutions are constructed by estimating orbital parameters for each 24-hour arc. The epochs of these daily arcs are recorded in the session tables. In the presence of an outlier, the affected epoch is removed, leading to the segmentation of the daily arc. New orbital parameters are then estimated for the revised arcs. As a result, while the general AIUB monthly gravity solution is typically based on daily arcs, months with outliers in the instrument exhibit shorter arcs in the monthly gravity solution.

P7L20: I suggest modification of this wording "a much looser outlier threshold" into "a much larger outlier threshold"

It was changed accordingly.

P8L1: Please correct: "3 shows" into "Figure 3 shows"

It was changed accordingly.

P10L4 and Figure 4: Your write "Then by plotting and 5 comparing the n gravity field solutions in terms of difference degree amplitudes of geoid heights …".  Please add here that you compare

with the monthly gravity field solutions ISTG-GRACE2018 and explain why you have chosen this TU Graz model for this comparison.

This paragraph was added:

To compute the difference degree amplitudes, selecting a reference gravity field solution is crucial. In this context, our choice is the monthly gravity field solutions produced by the Institute of Geodesy at Graz University of Technology, ITSG-GRACE2018. This decision is motivated by the intention to benchmark our solution against a high-quality GRACE gravity solution. Furthermore, our solution closely aligns with ITSG-GRACE2018 in terms of input observations, background models, and processing strategies. The deliberate use of a monthly model takes into account the varying quality of GRACE solutions from month to month, especially towards the end of the GRACE mission.

Please add the corresponding plot for Iteration 1 in figure 4.

Plot for iteration 1 in figure 4 was added.

P13L1-3 and Figure 6:   Yes, noise reduction of AIUB-G3P compared to AIUB-RL02 is obvious. But I do not really understand the "mean model". You state this model is simply the average of monthly gravity field solutions of several processing centres and you refer to Peter et al. 2022. But I'm sorry, I cannot find a description of such an average model in this paper. Or do you mean COST-G combined models? Please specify and explain also why you chose the "mean model".

This paragraph was added and the reference to Peter et al. 2022 was eliminated.

To maintain consistency, all comparisons in this section are referenced to a 'mean model'. The 'mean model' was computed by averaging monthly gravity field solutions from the Center for Space Research at the University of Texas, Austin (CSR Release 06), the German Research Centre for Geosciences (GFZ Release 06), the Centre National d'Etudes Spatiales/Groupe de Recherche de Ge\'ode\'sie Spatiale (CNES\_GRGS\_RL04) and the Institute of Geodesy at Graz University of Technology (ITSG-Grace2018) for the time period 2004-2017.

P14L5 and Figure 7: "Figure 7 shows 5 the noise over the ocean": Please describe in detail what you mean with "noise" resp. with "Weighted STD over the oceans", please specify what you computed. Please explain furthermore, why you did this computation over the oceans and not over land.

This paragraph was changed into:

One approach for evaluating GRACE gravity monthly solutions involves calculating the standard deviation (STD) of variability over the oceans, where hydrological signals are not expected. The discrepancies between the monthly solution and the 'mean model' are assessed on a grid with a cell size of 3 degrees, corresponding to a spherical harmonic expansion up to degree and order 60. Secular and seasonal variations are fitted to all grid cells and subtracted to eliminate long-periodic signals of oceanic origin. The grid cells are weighted by the cosine of the latitude to

account for their different sizes, and the standard deviation over all ocean cells is computed. To prevent contamination from continental signals, the shoreline is shifted by three grid cells (equivalent to 9 degrees) into the oceans. Figure 7 shows the standard deviations computed in this way for AIUB-RL02 and AIUB-G3P GRACE gravity field solutions.

P14L8 and Figure 8: "Figure 8 shows the RMS over the years spatially": Please describe also here what you computed. The caption of figure 8 also unclear for me: What are "RMS anomalies"? Has is something to do with gravity anomaly? In contrast, the color bar is indicated with "Temporal RMS in geoid height". What is "temporal RMS"?

Figure 8 has been excluded as its inclusion did not contribute any substantial content to the paper.

P15L3-6: The content of these sentences has already been explained previously in section 2

This paragraph was changed into:

As mentioned in Section 2, two significant changes in the processing chain have contributed to the improvement of AIUB-G3P:

Accelerometer parameterization: In AIUB-RL02, only the diagonal elements of the scale factor matrix were calculated for each accelerometer in each arc. In AIUB-G3P, the full scale factor matrix was computed for each accelerometer in each arc, aligning with the recommendations by \cite{klinger2016}.

Updating AOD release: The Atmospheric and Oceanic Dealiasing (AOD) release was updated from RL05 to RL06.

The impact of these two changes is assessed in Figure 8 for the years 2016-2017, a period during which the quality of GRACE observations was degraded.

P15L7-14 and Figure 9: Please describe also here what you mean with "noise over the oceans". I suppose this is the same functional as for figure 7.

The caption was changed into:

Weighted STD over the oceans for 2016 months where AIUB-RL02 are available.

P17L3ff and figure 11: This evaluation part is nice, but it's about gravity field solutions from JPL and TUG and has therefore no relation to the topic of your paper. Please remove this part.

Figure 11 was moved to the Appendix A.

---

## Author Response (AR2)

Thank you for your valuable comments; we appreciate your insightful feedback. Appendix A has been removed as per the request of Anonymous Referee #2.